# LATENT CAUSAL FOREST:
# ESTIMATING HETEROGENEOUS TREATMENT EFFECTS UNDER A HIDDEN TREATMENT

## ABSTRACT

Learning heterogeneous treatment effects from observational data is central to applications such as precision medicine and policy evaluation. A fundamental challenge arises when treatment assignment is unobserved and must instead be inferred from noisy proxies, inducing severe measurement error. We propose the Latent Causal Forest, a forest-based framework for estimating conditional average treatment effects (CATE) under a hidden treatment. The method combines the flexibility of forest models in capturing treatment effect heterogeneity with semiparametric efficiency guarantees via efficient influence functions. We establish consistency and asymptotic normality of the estimator and develop a two-step doubly robust learner tailored to the hidden-treatment setting. Simulation studies and empirical applications demonstrate that our approach outperforms existing alternatives. To our knowledge, this is the first framework that enables CATE estimation with a hidden binary treatment.

## 1 INTRODUCTION

Estimating conditional average treatment effects (CATE) has received increasing attention due to its central role in data-driven decision-making. In precision medicine, for instance, CATE estimation allows clinicians to recommend the right treatment to the right patient. A rich literature has developed machine learning methods for this task, including forests (Athey et al., 2019), boosting (Powers et al., 2017), meta-learners (Künzel et al., 2019), the *R-learner* (Nie & Wager, 2020), and the *DR-learner* (Kennedy, 2023), among others.

A key requirement of this literature is the correct specification of the treatment variable. In practice, however, treatment assignment is often *not directly observed*. This may arise due to misreporting or misallocation in assignment (Hu & Lewbel, 2012), or because the treatment itself is inherently latent (Þorsteinsson et al., 2021). For example, in biomedical sciences and health economics, treatment status is frequently self-reported, but patients may misstate their treatment, leading to biased effect estimates and ultimately suboptimal decisions.

Although there exists a substantial body of work on causal inference under measurement error, most methods focus on continuous treatments and marginal treatment effects (Schennach, 2016). By contrast, estimation of average or heterogeneous treatment effects under hidden binary treatments has received little attention, in part due to the non-additive nature of measurement error in this setting. Yet, uncovering heterogeneity is essential: it reveals how treatment effects vary across individuals and enables the design of optimal personalized treatment rules.

In this paper, we address this gap by developing methods for CATE estimation under hidden treatment. We introduce the **LatenT Causal Forest (LTCF)**, a forest-based framework that extends generalized random forests (Athey et al., 2019) to settings where the true treatment is unobserved but surrogate and proxy variables are available. Our approach leverages adaptive forest weights to capture treatment effect heterogeneity while incorporating semiparametric efficiency guarantees via influence-function adjustments. The method is multiply robust: the true heterogeneous treatment effect can be consistently recovered provided that at least one of several nuisance components is correctly specified. We establish consistency and asymptotic normality of the estimator, and further propose a two-step doubly robust learner tailored to the hidden-treatment setting.

By combining the flexibility of forests with the efficiency of semiparametric methods, our framework provides a robust and practical solution for CATE estimation when the treatment is latent. Empirical results on both synthetic and real-world data demonstrate that LTCF delivers substantial gains under hidden treatment.

## 1.1 RELATED WORK

**Measurement error with continuous treatment.** Much of the measurement error literature has focused on continuous treatments, typically under the assumption of additive error (Schennach, 2016). In this setting, the observed treatment is modeled as $A = A^* + \Delta A$, where $A^*$ is the true treatment and $\Delta A$ is an additive error. When the treatment–outcome relationship is linear, standard instrumental variables can correct for endogeneity (Amemiya, 1985). For nonlinear models, identification requires additional assumptions, such as known error distributions (Schennach, 2004), validation data (Hu & Ridder, 2010), or auxiliary variables (Hausman et al., 1991).

**Measurement error with binary treatment.** By contrast, research on binary treatments has been more limited, as misclassification errors are inherently nonadditive (Bollinger, 1996). Early approaches relied on validation datasets (Chen et al., 2005) or derived sharp bounds for the average treatment effect under nondifferential error assumptions (Imai & Yamamoto, 2010). Instrumental-variable methods have also been developed to address misclassification in regression models (Mahajan, 2006; DiTraglia & García-Jimeno, 2019). Most recently, Zhou & Tchetgen Tchetgen (2024) advanced this line of research by introducing the hidden treatment framework, which establishes identification and semiparametric estimation of the average treatment effect (ATE) when treatments are misclassified. Their approach leverages surrogate variables (error-prone measurements of the treatment) and proxy variables (exogenous sources of variation related to the true treatment), providing the first general methodology for ATE estimation under hidden binary treatment.

**Heterogeneous treatment effect.** Parallel to this work, a large literature has developed methods for conditional average treatment effect (CATE) estimation. Many approaches adopt two-step procedures that combine nuisance estimation with pseudo-outcome regression, such as the *R-learner* (Nie & Wager, 2020), *DR-learner* (Kennedy, 2023), and *mDR-learner* (Pryce et al., 2024). Our work instead builds on generalized random forests (GRF) (Athey et al., 2019), which view forests as adaptive locally weighted estimators capable of targeting general moment conditions. This framework ensures asymptotic consistency and normality while flexibly capturing treatment effect heterogeneity, and has been extended to survival analysis (Liu & Li, 2021; Cui et al., 2023) and to reduce sensitivity to nuisance estimation error (Oprescu et al., 2019).

## 2 PROBLEM STATEMENT

### 2.1 SETUP

We adopt the potential outcomes framework and assume the existence of potential outcomes $Y_i(1)$ and $Y_i(0)$, corresponding to the hidden binary treatment assignment $A_i^* \in \{0, 1\}$. Let $X_i \in \mathcal{X}$ denote a vector of pre-treatment covariates. The outcome variable $Y_i$ may be binary, count-based, or continuous. Our objective is to estimate the conditional average treatment effect (CATE), defined as

$$\tau(x) = E[Y_i(1) - Y_i(0) \mid X_i = x],$$

where $\tau(x)$ characterizes the treatment effect conditional on covariates $X_i = x$.

### 2.2 CAUSAL EFFECTS WITH OBSERVED TREATMENT

We first illustrate CATE estimation with the observed treatment $A^*$ under the framework of Causal Forests (Athey et al., 2019). Identification of $\tau(x)$ requires the following standard causal assumptions:

**Assumption 2.1 (Consistency)** $Y_i = Y_i(1)A^* + Y_i(0)(1 - A^*)$ *almost surely.*

**Assumption 2.2 (Overlap)** $0 < \Pr(A_i^* = a \mid X_i = x) < 1$ *for $a^* = 0, 1$ almost surely.*

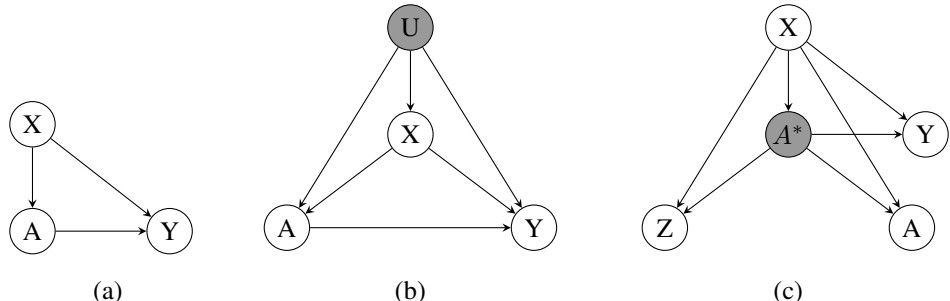

Figure 1: Illustration of causal graphs for different scenarios: (a) no unmeasured confounders, (b) with unobserved confounder $U$, and (c) with latent treatment $A^*$. Dark nodes indicate unobservability.

**Assumption 2.3 (Ignorability)** $\{Y_i(1), Y_i(0)\} \perp A_i^* \mid X_i$.

Under these assumptions, we can construct an efficient influence function (EIF)-based score to identify heterogeneous treatment effects in a nonparametric model. Denote $e_{a^*}(x) = \Pr(A_i^* = a^* \mid X_i = x)$, $m_{a^*}(x) = \mathbb{E}(Y_i \mid A_i^* = a^*, X_i = x)$. The score function is given by:

$$
\begin{aligned}
\psi_{\tau(x)}(X_i, Y_i, A_i^*; e_{a^*}, m_{a^*}) = & \frac{\mathbb{I}(A_i^* = 1)}{e_1(X_i)} \{Y_i - m_1(X_i)\} + m_1(X_i) \\
& - \frac{\mathbb{I}(A_i^* = 0)}{e_0(X_i)} \{Y_i - m_0(X_i)\} - m_0(X_i) - \tau(x).
\end{aligned}
\tag{1}
$$

This score function satisfies the doubly robust property: if either $e_{a^*}(x)$ or $m_{a^*}(x)$ is correctly specified, then $\mathbb{E}\left[\psi_{\tau(x)}(X_i, Y_i, A_i^*; e_{a^*}, m_{a^*}) \middle| X_i = x\right] = 0$. For categorical outcomes $Y$, the parameter of interest is $\mathbb{E}[\mathbb{I}(Y_i(1) = y) - \mathbb{I}(Y_i(0) = y) \mid X_i = x]$ where $y$ is a given level. Its score function can be obtained by replacing $Y$ with $\mathbb{I}(Y = y)$ in equation 1.

This framework also provides a convenient way to estimate the average treatment effect (ATE): $\tau = \mathbb{E}[\mathbb{E}[Y_i(1) - Y_i(0) \mid X_i]]$. We can first estimate the nuisance components $e_{a^*}(x)$ and $m_{a^*}(x)$ using machine learning methods, and then solve the following score equation to obtain a doubly robust estimator of the ATE:

$$
\sum_{i=1}^n \psi_{\hat{\tau}}(X_i, Y_i, A_i^*; \hat{e}_{a^*}, \hat{m}_{a^*}) = 0.
\tag{2}
$$

Our objective, however, is not limited to estimating a constant effect $\tau$, but to capture treatment heterogeneity as a function of covariates. To this end, we employ random forests to construct a weighted averaging estimator. For a test point $x$, the weights $\alpha_i(x)$—defined in Athey et al. (2019)—capture the frequency with which the $i$-th training observation falls into the same terminal leaf as $x$:

$$
\alpha_i(x) = \frac{1}{B} \sum_{b=1}^B \frac{\mathbb{I}\{X_i \in \mathcal{N}_b(x)\}}{|\mathcal{N}_b(x)|},
\tag{3}
$$

where $B$ is the total number of trees, $\mathcal{N}_b(x)$ is the terminal node containing $x$ in the $b$-th tree, and $|\cdot|$ denotes cardinality.

Forests are constructed to ensure that $\alpha_i(x)$ captures local treatment effect heterogeneity. Specifically, $\tau(x)$ is estimated by solving a localized version of equation 1:

$$
\sum_{i=1}^n \alpha_i(x) \psi_{\hat{\tau}(x)}(X_i, Y_i, A_i^*; \hat{e}_{a^*}, \hat{m}_{a^*}) = 0.
$$

## 3 CATE ESTIMATION WITH LTCF

In principle, Equation equation 3 provides a flexible approach to estimating heterogeneous treatment effects when treatment assignment is observed. However, when the treatment assignment is hidden, this approach breaks down because the nuisance parameters $e_{a^*}(x) = \Pr(A_i^* = a^* \mid X_i = x)$ and $m_{a^*}(x) = \mathbb{E}(Y_i \mid A_i^* = a^*, X_i = x)$ cannot be modeled directly, as $A_i^*$ is unobserved.

To address this challenge, we propose an approach that leverages a misclassified surrogate measure $A$. We then identify causal effects using the surrogate variable $A$ together with a proxy variable $Z$.

### 3.1 IDENTIFICATION WITH A HIDDEN TREATMENT

Suppose the true treatment status $A_i^*$ is not observed, and instead we observe a misclassified surrogate $A_i$. If one were to naively estimate treatment effects by substituting $A_i$ for $A_i^*$, the resulting latent outcome bias would be

$$\int \big[ \mathbb{E}(Y_i \mid A_i = a^*, X_i = x) - \mathbb{E}(Y_i \mid A_i^* = a^*, X_i = x) \big] dF(x),$$

which is generally nonzero.

To achieve correct identification of $\tau(x)$, we introduce the surrogate $A_i$ alongside a proxy variable $Z_i$, both related to the hidden treatment $A_i^*$. Identification requires the following assumptions.

**Assumption 3.1 (Relevance)** $Y_i \not\perp A_i^* \mid X_i;\ A_i \not\perp A_i^* \mid X_i;\ Z_i \not\perp A_i^* \mid X_i.$

**Assumption 3.2 (Conditional independence)** *$Y_i$, $A_i$, and $Z_i$ are mutually independent conditional on covariates and treatment assignment. That is $Y_i \perp A_i \perp Z_i \mid A_i^*, X_i$.*

**Assumption 3.3 (Surrogate)** *At least one of the following holds for any value $x$ of covariates:*

*(i)* $\Pr(A_i = 0 \mid A_i^* = 1, X_i = x) + \Pr(A_i = 1 \mid A_i^* = 0, X_i = x) < 1.$

*(ii)* $\Pr(A_i^* = 0 \mid A_i = 1, X_i = x) + \Pr(A^* = 1 \mid A_i = 0, X_i = x) < 1.$

*(iii)* $\Pr(A_i^* = 1 \mid A_i = 1, X_i = x) > \Pr(A_i^* = 0 \mid A_i = 1, X_i = x).$

*(iV)* $\Pr(A_i^* = 0 \mid A_i = 0, X_i = x) > \Pr(A_i^* = 1 \mid A_i = 0, X_i = x).$

Assumption 3.1 requires that, conditional on $X_i$, the hidden treatment $A_i^*$ is associated with the surrogate $A_i$ and the proxy $Z_i$. Assumption 3.2 requires that, conditional on $A_i^*$ and covariates, the surrogate and proxy are independent of the outcome $Y_i$. Assumption 3.3 ensures that $A_i$ contains sufficient information to recover $A_i^*$.

Figure 1 (c) illustrates the identification setting for causal effects with an unobserved binary treatment. This framework thus requires the presence of both the surrogate variable $A_i$ and the proxy variable $Z_i$, together with structural assumptions about their relationships. The figure also compares this case to scenarios with (a) no unmeasured confounders, (b) unmeasured confounders $U$.

**Lemma 3.4 (Identification)** *Under Assumptions 2.1-3.3, the heterogeneous treatment effects $\tau(x)$ can be uniquely identified.*

The proof of Lemma 3.4 is provided in Appendix B. Based on this result, $\tau(x)$ can be identified via the following EIF-based score function, motivated by semiparametric theory under hidden treatments and mapping from the oracle tangent space where $A_i^*$ is observed (Zhou & Tchetgen Tchetgen, 2024). Let $g_{a^*}(x) = \mathbb{E}(A_i \mid A_i^* = a^*, X_i = x), h_{a^*}(x) = \mathbb{E}(Z_i \mid A_i^* = a^*, X_i = x)$, and define the score function:

$$\psi_{\tau(x)}(X_i, Y_i, A_i, Z_i; g_{a^*}, h_{a^*}, e_{a^*}, m_{a^*})$$

$$= \frac{A_i - g_0(X_i)}{g_1(X_i) - g_0(X_i)} \frac{Z_i - h_0(X_i)}{h_1(X_i) - h_0(X_i)} \frac{1}{e_1(X_i)} \{Y_i - m_1(X_i)\} + m_1(X_i) - m_0(X_i)$$

$$- \frac{A_i - g_1(X_i)}{g_0(X_i) - g_1(X_i)} \frac{Z_i - h_1(X_i)}{h_0(X_i) - h_1(X_i)} \frac{1}{e_0(X_i)} \{Y_i - m_0(X_i)\} - \tau(x).$$

(4)

This EIF-based estimator enjoys a **triply robust property**:

$$\mathbb{E}\left[\psi_{\tau(x)}(X_i, Y_i, A_i, Z_i; g_{a^*}, h_{a^*}, e_{a^*}, m_{a^*})|X_i = x\right] = 0,$$

if any of the following conditions hold: (i) $g_{a^*}(x)$ and $m_{a^*}(x)$ are correctly specified; (ii) $h_{a^*}(x)$ and $m_{a^*}(x)$ are correctly specified; (iii) $g_{a^*}(x)$, $h_{a^*}(x)$ and $e_{a^*}(x)$ are correctly specified.

## 3.2 ESTIMATING CATE UNDER CAUSAL FOREST

The preceding theory provides a practical strategy for estimating the ATE under a hidden treatment. Identification of the ATE relies on the nuisance components $g_{a^*}(x)$, $h_{a^*}(x)$, $e_{a^*}(x)$, and $m_{a^*}(x)$. To capture heterogeneous effects, we develop a triply robust estimator for the conditional average treatment effect (CATE) using causal forests. Specifically, we employ causal forests to learn the weights and estimate $\hat{\tau}(x)$ via the moment equation:

$$\sum_{i=1}^{n} \alpha_i(x)\psi_{\hat{\tau}(x)}(X_i, Y_i, A_i, Z_i; \hat{g}_{a^*}, \hat{h}_{a^*}, \hat{e}_{a^*}, \hat{m}_{a^*}) = 0, \tag{5}$$

where $\psi_{\hat{\tau}(x)}$ is defined in Equation equation 4.

We now outline the forest-growing procedure. Following Athey et al. (2019), we adopt the $\widetilde{\Delta}$-criterion as the splitting rule. The goal is to partition a parent node $\mathcal{R}$ into two child nodes $\mathcal{R}_L$ and $\mathcal{R}_R$ within the subsample $\mathcal{J}$, so as to maximize the accuracy of $\tau$-estimates. This is formalized by minimizing the mean squared error (MSE). For node $\mathcal{R}$, the MSE of a split into $\mathcal{R}_L$ and $\mathcal{R}_R$ is

$$\text{MSE}\left(\mathcal{R}_L, \mathcal{R}_R\right) = \left(\sum_{i \in \mathcal{R}_L} \left(\tau(x_i) - \hat{\tau}(\mathcal{R}_L)\right)^2 + \sum_{i \in \mathcal{R}_R} \left(\tau(x_i) - \hat{\tau}(\mathcal{R}_R)\right)^2\right),$$

where $\hat{\tau}(\mathcal{R})$ is the solution to equation 2 using the score function equation 4 within node $\mathcal{R}$. Since $\tau(x)$ is unobserved, the MSE is infeasible. Instead, we rely on an abstract target criterion. Denote $n_{\mathcal{R}} = |\{i \in \mathcal{J} : X_i \in R\}|$ and $n_{\mathcal{R}_L}, n_{\mathcal{R}_R}$ as the sample sizes of the parent and child nodes. Then,

$$\Delta(\mathcal{R}_L, \mathcal{R}_R) = \frac{n_{\mathcal{R}_L} n_{\mathcal{R}_R}}{n_{\mathcal{R}}^2} \left(\hat{\tau}(\mathcal{R}_L) - \hat{\tau}(\mathcal{R}_R)\right)^2.$$

Assuming $n_{\mathcal{R}_L}, n_{\mathcal{R}_R} \gg r^{-2}$ for parent node radius $r > 0$, we have

$$\mathbb{E}\left[\text{MSE}\left(\mathcal{R}_L, \mathcal{R}_R\right)\right] = K(\mathcal{R}) - \mathbb{E}\left[\Delta(\mathcal{R}_L, \mathcal{R}_R)\right] + o\left(r^2\right),$$

where $K(\mathcal{R})$ is a constant reflecting the purity of the parent node, and $o(r^2)$ captures sampling variation. Direct computation of $\tau(\mathcal{R}_L)$ and $\tau(\mathcal{R}_R)$ for all candidate splits is computationally intensive and complex. To address this, an equivalent estimator is used to reduce computational burden without compromising accuracy:

$$\widetilde{\Delta}(\mathcal{R}_L, \mathcal{R}_R) = \frac{1}{n_{\mathcal{R}_L}}\left(\sum_{i \in \mathcal{R}_L} \psi^i_{\hat{\tau}(\mathcal{R})}\right)^2 + \frac{1}{n_{\mathcal{R}_R}}\left(\sum_{i \in \mathcal{R}_R} \psi^i_{\hat{\tau}(\mathcal{R})}\right)^2,$$

with $\psi^i_{\hat{\tau}(\mathcal{R})} = \psi_{\hat{\tau}(\mathcal{R})}(X_i, Y_i, A_i, Z_i; \hat{g}_{a^*}, \hat{h}_{a^*}, \hat{e}_{a^*}, \hat{m}_{a^*})$. Since $\widetilde{\Delta}(\mathcal{R}_L, \mathcal{R}_R)$ is equivalent to $\Delta(\mathcal{R}_L, \mathcal{R}_R)$ (by Proposition 1 of Athey et al. (2019)), splits are chosen to maximize $\widetilde{\Delta}(\mathcal{R}_L, \mathcal{R}_R)$.

## 3.3 LEARNING NUISANCE COMPONENTS

A key element of LTCF is estimating the nuisance functions $g_{a^*}(x)$, $h_{a^*}(x)$, $e_{a^*}(x)$, and $m_{a^*}(x)$. This is challenging because the true treatment $A_i^*$ is unobserved. Building on Tatiana Benaglia & Hunter (2009), one strategy uses an Expectation-Maximization (EM)–based method for estimating "posterior" probabilities of latent components.

We adopt the kernel regression estimator of Zhou & Tchetgen Tchetgen (2024), which extends this approach by constructing "posterior" probabilities for unobserved $A_i^*$ rather than mixture components, and introduces Gaussian kernel regression estimators for $g_{a^*}(x)$, $h_{a^*}(x)$, $e_{a^*}(x)$, and $m_{a^*}(x)$. To further enhance robustness and mitigate overfitting, we incorporate cross-fitting (Schick, 1986a), whereby nuisance functions are trained on one fold and then applied to construct score functions for another, ensuring valid CATE estimation.

## 3.4 THEORETICAL ANALYSIS

In this section, we establish the theoretical properties of the proposed LTCF estimator. Let $\tilde{\tau}(x)$ denote the oracle estimator of $\tau(x)$, and let $\hat{\tau}(x)$ denote the LTCF estimator obtained by replacing the nuisance components $g_{a^*}(x), h_{a^*}(x), e_{a^*}(x), m_{a^*}(x)$ with their estimated counterparts.

**Assumption 3.5 (Consistency of nuisance components)** *For $a^* \in \{0, 1\}$,*

$$\mathbb{E}\left[\sup_{x \in \mathcal{X}} |\hat{e}_{a^*}(x) - e_{a^*}(x)|^2\right] = o(e_n^2), \mathbb{E}\left[\sup_{x \in \mathcal{X}} |\hat{m}_{a^*}(x) - m_{a^*}(x)|^2\right] = o(m_n^2),$$

$$\mathbb{E}\left[\sup_{x \in \mathcal{X}} |\hat{g}_{a^*}(x) - g_{a^*}(x)|^2\right] = o(g_n^2), \mathbb{E}\left[\sup_{x \in \mathcal{X}} |\hat{h}_{a^*}(x) - h_{a^*}(x)|^2\right] = o(h_n^2).$$

Under Assumption 3.5, we obtain the following intermediate result, which bounds the difference between the estimated and oracle CATE.

**Lemma 3.6** *If Assumption 3.5 holds, then for any $x \in \mathcal{X}$,*

$$\hat{\tau}(x) - \tilde{\tau}(x) = o_p(\max(m_n g_n, m_n h_n, m_n e_n, g_n h_n)).$$

The proof of Lemma 3.6 is provided in Appendix C. Following Athey et al. (2019), our forest construction assumes honesty and standard regularity conditions: at each internal node, the probability of splitting on any feature is at least $\varsigma > 0$, and each split allocates at least a fraction $\omega > 0$ of observations to both child nodes. Trees are grown on subsamples of size $s = n^\beta$ with $\beta_{\min} < \beta < 1$, where

$$\beta_{\min} := 1 - \left(1 + \pi^{-1} \left(\log\left(\omega^{-1}\right)\right) / \left(\log\left((1-\omega)^{-1}\right)\right)^{-1}\right), \tag{6}$$

$\pi$ is the minimum probability that the tree splits on any feature, and covariates $X \in [0,1]^p$ follow a density bounded away from zero and infinity.

**Assumption 3.7 (Lipschitz continuity)** *The treatment effect $\tau(x)$ and nuisance functions $g_{a^*}(x), h_{a^*}(x), e_{a^*}(x), m_{a^*}(x)$ are Lipschitz continuous in $x$.*

Combining Lemma 3.6 with the results of Athey et al. (2019), we establish the following asymptotic normality result.

**Theorem 3.8** *Suppose Assumptions 2.1–3.3, 3.5, and 3.7 hold, and $s$ scales as in equation 6. If $o_p(\max(m_n g_n, m_n h_n, m_n e_n, g_n h_n))$ converges to zero at a rate faster than $polylog(n/s)^{-1/2}(s/n)^{1/2}$, then for any $x \in \mathcal{X}$,*

$$[\hat{\tau}(x) - \tau(x)]/\sigma_n(x) \to N(0, 1),$$

*where $\sigma_n^2(x) = polylog(n/s)^{-1} \cdot (s/n)$ is bounded away from zero and grows at most polynomially in $\log(n/s)$.*

The proof of Theorem 3.8 is given in Appendix D. To estimate the asymptotic variance of $\hat{\tau}(x)$, we adopt the "bootstrap of little bags" approach (Sexton & Laake, 2009; Athey et al., 2019). Specifically, we use

$$\tilde{\tau}^*(x) = \tau(x) + \sum_{i=1}^n \alpha_i(x) \psi_{\tau(x)}^i,$$

which is asymptotically equivalent to $\hat{\tau}(x)$ under Theorem 3.8. Consequently,

$$\sigma_n^2(x) = \text{Var}\left[\tilde{\tau}^*(x)\right] = \text{Var}\left[\sum_{i=1}^n \alpha_i(x) \psi_{\tau(x)}^i\right].$$

We estimate $\sigma_n^2(x)$ using the bootstrap of little bags, which accounts for within- and between-tree correlations and provides a computationally efficient approximation to the full bootstrap variance.

### 3.5 DR-LEARNER WITH HIDDEN TREATMENT

We now present an alternative strategy for CATE estimation under latent treatment, adapting the *DR-learner* framework of Kennedy (2023). The DR-learner separates nuisance estimation from CATE estimation in two stages: (i) estimating nuisance components to construct pseudo-outcomes, and (ii) modeling treatment effects using flexible learners. Cross-validation is employed for model selection and tuning, and finite-sample error bounds are available relative to an oracle benchmark. Related extensions include the *P-learner* (Sverdrup & Cui, 2023), which adapts this strategy to proximal causal inference.

Building on these ideas, we propose a two-stage *triply robust* DR-learner specifically tailored for a hidden treatment, under Assumptions 2.1–3.3. For the average treatment effect (ATE), $\mathbb{E}[Y(1) - Y(0)]$, we define the following triply robust score, motivated by semiparametric efficiency theory (Zhou & Tchetgen Tchetgen, 2024):

$$
\begin{aligned}
\Gamma_i = {} & \frac{A_i - g_0(X_i)}{g_1(X_i) - g_0(X_i)} \frac{Z_i - h_0(X_i)}{h_1(X_i) - h_0(X_i)} \frac{1}{e_1(X_i)} \{Y_i - m_1(X_i)\} \\
& - \frac{A_i - g_1(X_i)}{g_0(X_i) - g_1(X_i)} \frac{Z_i - h_1(X_i)}{h_0^*(X_i) - h_1(X_i)} \frac{1}{e_0(X_i)} \{Y_i - m_0(X_i)\} \\
& + m_1(X_i) - m_0(X_i).
\end{aligned}
\tag{7}
$$

The sample average of $\Gamma_i$ yields a triply robust estimator of the ATE.

To estimate the CATE, we regress the pseudo-outcome $\Gamma_i$ on $X_i$. Specifically, the final-stage model is given by

$$
\hat{\tau}_{dr}(x) = \mathbb{E}\left[ \hat{\Gamma}_i^{(-c(i))} | X_i = x \right],
$$

where cross-fitting ensures robustness to nuisance estimation errors. This predictive problem is highly flexible and can be addressed using nonparametric learners (e.g., random forests, neural networks, stacking) or simpler parametric models. The detailed algorithm is provided in Appendix A.

While this two-stage procedure inherits desirable oracle properties (Kennedy, 2023), valid pointwise inference for $\hat{\tau}(x)$ remains challenging when nuisance estimates are imperfect. Nevertheless, the triply robust DR-learner offers a practical and theoretically grounded approach for CATE estimation under hidden treatment.

## 4 NUMERICAL EXPERIMENTS

### 4.1 SYNTHETIC SCENARIOS

We generate synthetic data following the auxiliary variable mechanism of Zhou & Tchetgen Tchetgen (2024), where treatment heterogeneity is incorporated through a CATE function for both binary and continuous outcomes (a detailed setup is provided in Appendix E). Cross-fitting (Chernozhukov et al., 2018; Schick, 1986b) is used to estimate the nuisance components $g_{a^*}$, $h_{a^*}$, $e_{a^*}$, and $m_{a^*}$. Since no existing methods are designed specifically for CATE estimation under hidden treatment, we benchmark against surrogate approaches with surrogate $A$ and proxy $Z$ as treatments. The tuning parameters are described in Appendix E. A test dataset, equal in size to the training set, is used to evaluate LTCF and surrogate causal forests. We measure the accuracy of estimated heterogeneous treatment effects using Mean squared error (MSE), which is defined as $\frac{1}{n_{test}} \sum_{i=1}^{n_{test}} (\hat{\tau}(X_i) - \tau(X_i))^2$.

Figure 2a reports the simulation results. In each panel, the horizontal axis shows the true CATE and the vertical axis the estimated CATE; hence, accuracy is indicated by closeness to the 45-degree line. We compare our method, LTCF, with three alternatives: (i) Oracle LTCF, which uses ground-truth nuisance parameters; (ii) Surrogate CF, which incorrectly uses the surrogate $A$ as the treatment; and (iii) Proxy CF, which incorrectly uses the proxy $Z$ as the treatment. The results show that LTCF closely aligns with the oracle benchmark, while treating $A$ or $Z$ as the treatment induces substantial bias.

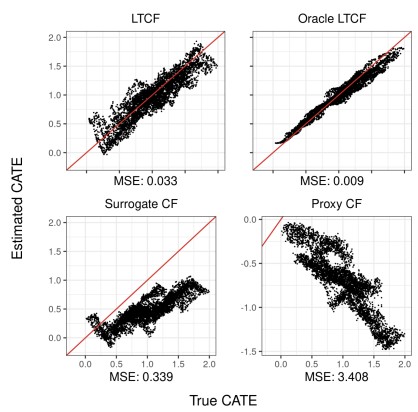 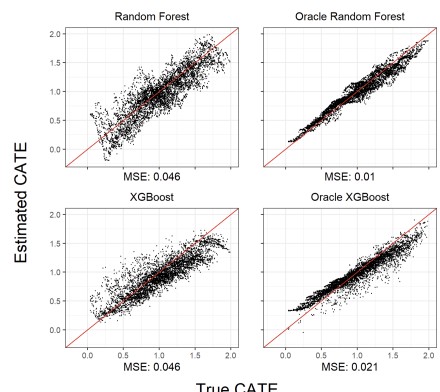

(a) Simulation results under LTCF with continuous outcomes. Top: LTCF and oracle LTCF. Bottom: causal forest with surrogate $A$ and proxy $Z$ as treatments.

(b) Simulation results for the DR-learner with regression forest (top) and XGBoost (bottom). Left: estimated vs. true CATEs with estimated nuisances. Right: same with Oracle nuisances.

Figure 2: Empirical evaluation of CATE estimation under a hidden treatment. (a) LTCF versus benchmarks in continuous outcome settings. (b) DR-learner with regression forest and XGBoost implementations.

Table 1: Mean and standard deviation of MSE (scaled by 100) for $n = 2000, 5000, 8000$ in LTCF and surrogate causal forests. Results are averaged over 200 replications.

|  | $n$ | **Surrogate CF** | **LTCF** |
|---|---|---|---|
|  | 2000 | 3.82(0.79) | 1.19(0.32) |
| Binary | 5000 | 3.87(0.57) | 0.78(0.18) |
|  | 8000 | 3.75(0.42) | 0.70(0.14) |
|  | 2000 | 35.42(5.88) | 6.92(1.97) |
| Continuous | 5000 | 34.42(3.81) | 6.01(1.13) |
|  | 8000 | 34.59(3.07) | 4.45(0.63) |

Table 1 further summarizes the mean squared error relative to the true CATE $\tau(x)$, averaged over 200 replications. The presence of measurement error in surrogate $A_i$ introduces severe bias in Surrogate CF, resulting in substantially higher MSE. By contrast, LTCF achieves much lower error across all sample sizes. Table 2, reports the coverage of 95% confidence intervals at four deterministic points, $x_1 = (0.2, 0.2)^T$, $x_2 = (0.4, 0.4)^T$, $x_3 = (0.6, 0.6)^T$, and $x_4 = (0.8, 0.8)^T$, demonstrating reliable coverage performance of the proposed intervals.

We also evaluate the proposed DR-learner with hidden treatment (Section 3.5) under the same continuous-outcome setting, using regression forests (Athey et al., 2019) and XGBoost (Chen & Guestrin, 2016) as base learners. Tuning parameters are selected by cross-validation. As shown in

Table 2: Coverage probability (%) of the proposed 95% confidence intervals by LTCF at four deterministic points. The numbers are aggregated over 200 simulation replications.

|  | $n$ | $x_1$ | $x_2$ | $x_3$ | $x_4$ |
|---|---|---|---|---|---|
|  | 2000 | 95.15 | 95.15 | 93.20 | 96.12 |
| Binary | 5000 | 90.03 | 91.32 | 93.10 | 96.55 |
|  | 8000 | 95.00 | 97.14 | 95.00 | 93.57 |
|  | 2000 | 91.80 | 93.44 | 93.44 | 94.26 |
| Continuous | 5000 | 95.33 | 95.33 | 91.59 | 89.72 |
|  | 8000 | 97.14 | 92.86 | 92.86 | 94.29 |

Figure 2b, both implementations achieve good performance, with XGBoost exhibiting noticeably higher accuracy than regression forests, reflecting its superior learning capacity.

## 4.2 REAL DATA APPLICATION

We apply the proposed methods to data from the Alzheimer's Disease Neuroimaging Initiative (ADNI). Our focus is on the effect of Alzheimer's disease (AD) on hippocampal volume. Diagnoses of AD may not perfectly reflect the latent disease status due to the complexity of the condition. To address this, we use the recall subscore of the Mini-Mental State Examination (MMSE) as a surrogate variable ($A$) and the orientation subscore as a proxy variable ($Z$), following Zhou & Tchetgen Tchetgen (2024).

We estimate the CATE of AD on hippocampal atrophy, where the hidden treatment is AD status at Month 6 and the outcome is hippocampal volume at Month 12. Covariates $X$ include age, years of education, and gender. The dataset comprises 1,176 subjects with complete MMSE scores at Month 6 and hippocampal volume measurements at Month 12. To examine heterogeneity, we estimate subgroup effects by gender.

Our results indicate that AD reduces hippocampal volume by 897.7-2150.1 $mm^3$ in men (average effect: 1533.4 $mm^3$), and by 503.3 -1763.7 $mm^3$ in women (average effect: 1253.3 $mm^3$). These findings suggest that women experience greater overall hippocampal atrophy due to AD compared to men, consistent with prior evidence of accelerated atrophy in female patients. To further validate our results, we assess performance using the Targeting Operator Characteristic (TOC); details are provided in Appendix F. This result indicates the presence of heterogeneity in response to the right AD status. For example, as shown in Figure 3, patients in the lowest 20th quartile of estimated CATEs have hippocampal volumes that are, on average, 400 $mm^3$ lower than those of the overall population when in AD status. This indicates a significantly more detrimental effect on brain composition for this subgroup compared to whole population

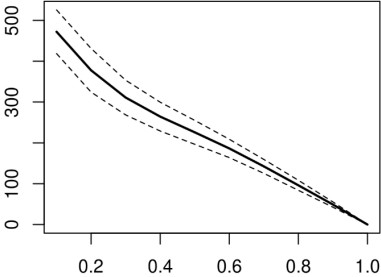

Figure 3: The TOC curve for all observations estimated by LTCF.

## 5 DISCUSSION

A central approach to assessing treatment effect heterogeneity is the estimation of CATE. In many applications, however, the treatment variable is not directly observed. We proposed a new framework for CATE estimation with unobserved treatments, based on pseudo-outcome splitting and forest-based weighting, which achieves consistency and asymptotic normality under reasonably accurate nuisance estimation. We also extended the *DR-learner* (Kennedy, 2023) to the hidden treatment setting, demonstrating its viability in practice.

A natural extension is to estimate CATEs conditional on both $A_i$ and $Z_i$, defined as $\tau(x, z) = \mathbb{E}\left[Y_i(1) - Y_i(0) \mid X_i = x, Z_i = z\right]$ and $\tau(x, a) = \mathbb{E}\left[Y_i(1) - Y_i(0) \mid X_i = x, A_i = a\right]$. Moreover, identification in our hidden treatment framework relies on the ignorability of the latent treatment $A^*$. An important direction for future research is to develop identification strategies that remain valid when this ignorability assumption is violated.

ETHICS STATEMENT

This study is based on simulated data and publicly available datasets. No new data involving human participants or animals were collected. All analyses adhere to ethical guidelines for research integrity and transparency. The authors confirm that the study design, implementation, and reporting comply with established standards of responsible research practice.

REPRODUCIBILITY STATEMENT

Code and scripts are provided in the supplementary materials to replicate the empirical results.

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

## A  ALGORITHMS OF THE DR-LEARNER

---

**Algorithm 1** DR-learner under Hidden Treatment

---

**Step 1**: Split the data $i = 1, \ldots, n$ into $C$ evenly sized folds. Estimate $g_{a^*}(x)$, $h_{a^*}(x)$, $e_{a^*}(x)$, and $m_{a^*}(x)$ using cross-fitting over the $C$ folds.

**Step 2**: Form the scores equation 7 using cross-fitted plug-in estimates $\hat{g}_{a^*}^{-c(i)}(x)$, $\hat{h}_{a^*}^{-c(i)}(x)$, $\hat{e}_{a^*}^{-c(i)}(x)$, and $\hat{m}_{a^*}^{-c(i)}(x)$, where $c(\cdot)$ maps each sample to its fold, and $-c(i)$ indicates predictions obtained without using the fold containing the $i$-th sample.

Estimate treatment effects by minimizing the empirical loss

$$\hat{\tau}(\cdot) = \arg\min_{\tau} \left[ \hat{L}_n(\tau(\cdot)) \right],$$

where

$$\hat{L}_n(\tau(\cdot)) = \tfrac{1}{n} \sum_{i=1}^{n} \left( \hat{\Gamma}_i^{(-c(i))} - \tau(X_i) \right)^2,$$

and $\hat{\Gamma}_i^{(-c(i))}$ are cross-fitted estimates of the scores equation 7, obtained by replacing $g_{a^*}(x)$, $h_{a^*}(x)$, $e_{a^*}(x)$, and $m_{a^*}(x)$ with $\hat{g}_{a^*}^{-c(i)}(x)$, $\hat{h}_{a^*}^{-c(i)}(x)$, $\hat{e}_{a^*}^{-c(i)}(x)$, and $\hat{m}_{a^*}^{-c(i)}(x)$, respectively.

---

## B  PROOF OF LEMMA 3.4

By Assumptions 2.1, 2.2, and 2.3, estimating $E[Y(a) \mid X = x]$ reduces to estimating $E[Y \mid X = x, A^* = a]$.

Suppose we observe data $\{X_i, Y_i, Z_i, A_i\}$; for simplicity, we omit the index $i$. Under Assumption 3.2,

$$f(A, Z \mid X = x) = \sum_{a^*=0}^{1} f(A, Z \mid X = x, A^* = a^*) f(A^* = a^* \mid X = x)$$

$$= \sum_{a^*=0}^{1} f(A \mid X = x, A^* = a^*) f(A^* = a^* \mid X = x) f(Z \mid X = x, A^* = a^*).$$

Similarly,

$$f(Y, Z, A \mid X = x)$$

$$= \sum_{a^*=0}^{1} f(Y, Z, A \mid X = x, A^* = a^*) f(A^* = a^* \mid X = x)$$

$$= \sum_{a^*=0}^{1} f(Y \mid X = x, A^* = a^*) f(A^* = a^* \mid X = x) f(A \mid X = x, A^* = a^*) f(Z \mid X = x, A^* = a^*).$$

By Assumption 3.1, $f(A, Z \mid X = x)$ is invertible. Define

$$\boldsymbol{P}(A, Z \mid X = x) := \left( f(A = 2 - i, Z = 2 - j \mid X = x) \right)_{i,j},$$

$$\boldsymbol{P}(Y, Z, A \mid X = x) := \left( f(Y, A = 2 - i, Z = 2 - j \mid X = x) \right)_{i,j},$$

$$\boldsymbol{P}(A \mid X = x, A^*) := \left( f(A = 2 - i \mid X = x, A^* = 2 - j) \right)_{i,j},$$

$$\boldsymbol{P}(Z \mid X = x, A^*) := \left( f(Z = 2 - i \mid X = x, A^* = 2 - j) \right)_{i,j},$$

$$\boldsymbol{P}(Y \mid X = x, A^*) := \operatorname{diag}\left( f(Y \mid X = x, A^* = 2 - i) \right),$$

$$\boldsymbol{P}(A^* \mid X = x) := \operatorname{diag}\left( f(A^* = 2 - i \mid X = x) \right)_{i},$$

where $i, j \in \{1, 2\}$.

Then the above equations can be rewritten as

$$\boldsymbol{P}(A, Z \mid X = x) = \boldsymbol{P}(A \mid X = x, A^*) \, \boldsymbol{P}(A^* \mid X = x) \, \boldsymbol{P}(Z \mid X = x, A^*)^{\top},$$

$$\boldsymbol{P}(Y, A, Z \mid X = x) = \boldsymbol{P}(A \mid X = x, A^*) \, \boldsymbol{P}(Y \mid X = x, A^*) \, \boldsymbol{P}(A^* \mid X = x) \, \boldsymbol{P}(Z \mid X = x, A^*)^{\top}.$$

Multiplying $\boldsymbol{P}(Y, A, Z \mid X = x)$ by $\boldsymbol{P}(A, Z \mid X = x)^{-1}$ yields

$$\boldsymbol{P}(Y, A, Z \mid X = x)\, \boldsymbol{P}(A, Z \mid X = x)^{-1} = \boldsymbol{P}(A \mid X = x, A^*)\, \boldsymbol{P}(Y \mid X = x, A^*)\, \boldsymbol{P}(A \mid X = x, A^*)^{-1}.$$

This is the canonical form of matrix eigen-decomposition. By Assumption 3.3, $\boldsymbol{P}(Y \mid X = x, A^*)$ can be solved as the eigenvalues. Therefore, the treatment effect $E[Y \mid X = x, A^* = 1] - E[Y \mid X = x, A^* = 0]$ can be uniquely identified from the data.

## C  PROOF OF LEMMA 3.6

We begin by proving the result in the context of estimating

$$\mu_{a^*}(x) = \mathbb{E}(Y_i \mid A_i^* = a^*, X_i = x),$$

and show that

$$\hat{\mu}_{a^*}(x) - \tilde{\mu}_{a^*}(x) = o_p\big(\max(m_n g_n, m_n h_n, m_n e_n, g_n h_n)\big).$$

Cross-fitting employs cross-fold estimation to mitigate bias due to overfitting. Specifically, observations $I$ that fall into the same terminal node as $x$ are randomly split into two subsets, $I_1$ and $I_2$. The estimator is then defined as

$$\hat{\mu}_{a^*}(x) = \frac{l_1}{l}\hat{\mu}_{a^*}^{I_1}(x) + \frac{l_2}{l}\hat{\mu}_{a^*}^{I_2}(x),$$

where $l = |I|$, $l_1 = |I_1|$, $l_2 = |I_2|$, and $\hat{\mu}_{a^*}^{I_1}(x), \hat{\mu}_{a^*}^{I_2}(x)$ are constructed using nuisance functions estimated from the complementary samples $I_2$ and $I_1$, respectively.

We aim to show that

$$\hat{\mu}_{a^*}^{I_1}(x) - \tilde{\mu}_{a^*}^{I_1}(x) = o_p\big(\max(m_n g_n, m_n h_n, m_n e_n, g_n h_n)\big),$$

where $\tilde{\mu}_{a^*}^{I_1}(x)$ is the oracle estimator of $\mu_{a^*}^{I_1}(x)$ obtained by solving

$$\frac{1}{l_1}\sum_{i:i\in I_1}\Big(\frac{A_i - g_{1-a^*}(X_i)}{g_{a^*}(X_i) - g_{1-a^*}(X_i)}\frac{Z_i - h_{1-a^*}(X_i)}{h_{a^*}(X_i) - h_{1-a^*}(X_i)}\frac{1}{e_{a^*}(X_i)}\{Y_i - m_{a^*}(X_i)\}$$
$$+ m_{a^*}(X_i) - \mu_{a^*}^{I_1}(x)\Big) = 0.$$

We now decompose $\hat{\mu}_{a^*}^{I_1}(x) - \tilde{\mu}_{a^*}^{I_1}(x)$ :

$$\hat{\mu}_{a^*}^{I_1}(x) - \tilde{\mu}_{a^*}^{I_1}(x)$$

$$= \frac{1}{l_1}\left[\sum_{i\in I_1}\frac{A_i - \hat{g}_{1-a^*}(X_i)}{\hat{g}_{a^*}(X_i) - \hat{g}_{1-a^*}(X_i)}\frac{Z_i - \hat{h}_{1-a^*}(X_i)}{\hat{h}_{a^*}(X_i) - \hat{h}_{1-a^*}(X_i)}\frac{1}{\hat{e}_{a^*}(X_i)}\big(Y_i - \hat{m}_{a^*}(X_i)\big) + \hat{m}_{a^*}(X_i)\right]$$

$$- \frac{1}{l_1}\left[\sum_{i\in I_1}\frac{A_i - g_{1-a^*}(X_i)}{g_{a^*}(X_i) - g_{1-a^*}(X_i)}\frac{Z_i - h_{1-a^*}(X_i)}{h_{a^*}(X_i) - h_{1-a^*}(X_i)}\frac{1}{e_{a^*}(X_i)}\big(Y_i - m_{a^*}(X_i)\big) + m_{a^*}(X_i)\right]$$

$$= \frac{1}{l_1}\left[\sum_{i\in I_1}\frac{A_i - g_{1-a^*}(X_i)}{g_{a^*}(X_i) - g_{1-a^*}(X_i)}\frac{Z_i - h_{1-a^*}(X_i)}{h_{a^*}(X_i) - h_{1-a^*}(X_i)}\frac{1}{e_{a^*}(X_i)}\big(m_{a^*}(X_i) - \hat{m}_{a^*}(X_i)\big)\right.$$

$$+ \hat{m}_{a^*}(X_i) - m_{a^*}(X_i)$$

$$+ \left(\frac{A_i - \hat{g}_{1-a^*}(X_i)}{\hat{g}_{a^*}(X_i) - \hat{g}_{1-a^*}(X_i)}\frac{Z_i - \hat{h}_{1-a^*}(X_i)}{\hat{h}_{a^*}(X_i) - \hat{h}_{1-a^*}(X_i)}\frac{1}{\hat{e}_{a^*}(X_i)}\right.$$

$$\left.- \frac{A_i - g_{1-a^*}(X_i)}{g_{a^*}(X_i) - g_{1-a^*}(X_i)}\frac{Z_i - h_{1-a^*}(X_i)}{h_{a^*}(X_i) - h_{1-a^*}(X_i)}\frac{1}{e_{a^*}(X_i)}\right)\big(Y_i - m_{a^*}(X_i)\big)$$

$$+ \left(\frac{A_i - \hat{g}_{1-a^*}(X_i)}{\hat{g}_{a^*}(X_i) - \hat{g}_{1-a^*}(X_i)}\frac{Z_i - \hat{h}_{1-a^*}(X_i)}{\hat{h}_{a^*}(X_i) - \hat{h}_{1-a^*}(X_i)}\frac{1}{\hat{e}_{a^*}(X_i)}\right.$$

$$\left.\left.- \frac{A_i - g_{1-a^*}(X_i)}{g_{a^*}(X_i) - g_{1-a^*}(X_i)}\frac{Z_i - h_{1-a^*}(X_i)}{h_{a^*}(X_i) - h_{1-a^*}(X_i)}\frac{1}{e_{a^*}(X_i)}\right)\big(m_{a^*}(X_i) - \hat{m}_{a^*}(X_i)\big)\right]. \tag{8}$$

Denote $\hat{\mu}_{a^*}^{I_1}(x) - \tilde{\mu}_{a^*}^{I_1}(x)$ by $K_1 + K_2 + K_3$.

**Analysis of $K_1$.** By the triply robust property of equation 4,

$$\mathbb{E}\left[\frac{A_i - g_{1-a^*}(X_i)}{g_{a^*}(X_i) - g_{1-a^*}(X_i)}\frac{Z_i - h_{1-a^*}(X_i)}{h_{a^*}(X_i) - h_{1-a^*}(X_i)}\frac{1}{e_{a^*}(X_i)}\big(m_{a^*}(X_i) - \hat{m}_{a^*}(X_i)\big)\right.$$

$$\left.+ \hat{m}_{a^*}(X_i) - m_{a^*}(X_i)\right] = 0.$$

Conditioned on $X_i = x$, we have

$$\mathbb{E}\left[\frac{A_i - g_{1-a^*}(X_i)}{g_{a^*}(X_i) - g_{1-a^*}(X_i)}\frac{Z_i - h_{1-a^*}(X_i)}{h_{a^*}(X_i) - h_{1-a^*}(X_i)} \mid X_i = x\right] = e_{a^*}(x_i).$$

Let $\Delta_{m_i} = m_{a^*}(X_i) - \hat{m}_{a^*}(X_i)$. Then

$$\mathbb{E}\left[\frac{A_i - g_{1-a^*}(X_i)}{g_{a^*}(X_i) - g_{1-a^*}(X_i)}\frac{Z_i - h_{1-a^*}(X_i)}{h_{a^*}(X_i) - h_{1-a^*}(X_i)}\frac{1}{e_{a^*}(X_i)}\big(\Delta_{m_i}\big) - \Delta_{m_i}\right]$$

$$= \mathbb{E}\left[\mathbb{E}\left[\frac{A_i - g_{1-a^*}(x)}{g_{a^*}(x) - g_{1-a^*}(x)}\frac{Z_i - h_{1-a^*}(x)}{h_{a^*}(x) - h_{1-a^*}(x)}\frac{1}{e_{a^*}(x)}\big(\Delta_{m_i}\big) - \Delta_{m_i}|X_i = x\right]\right]$$

$$= 0.$$

Thanks to cross-fitting, nuisance components can be treated as deterministic. Thus, conditional on $I_2$, the summands are mean-zero and independent, leading to the variance bound in equation 9, which is $o_p(1/l)$.

$$\mathbb{E}\left[\left(\frac{1}{l_1}\sum_{i\in I_1}\frac{A_i-g_{1-a^*}(X_i)}{g_{a^*}(X_i)-g_{1-a^*}(X_i)}\frac{Z_i-h_{1-a^*}(X_i)}{h_{a^*}(X_i)-h_{1-a^*}(X_i)}\frac{1}{e_{a^*}(X_i)}\big(m_{a^*}(X_i)-\hat{m}_{a^*}(X_i)\big)\right.\right.$$

$$\left.\left.+\,\hat{m}_{a^*}(X_i)-m_{a^*}(X_i)\right)^2\right]$$

$$=\mathbb{E}\left[\mathbb{E}\left[\left(\frac{1}{l_1}\sum_{i\in I_1}\frac{A_i-g_{1-a^*}(X_i)}{g_{a^*}(X_i)-g_{1-a^*}(X_i)}\frac{Z_i-h_{1-a^*}(X_i)}{h_{a^*}(X_i)-h_{1-a^*}(X_i)}\frac{1}{e_{a^*}(X_i)}\big(m_{a^*}(X_i)-\hat{m}_{a^*}(X_i)\big)\right.\right.\right.$$

$$\left.\left.\left.+\,\hat{m}_{a^*}(X_i)-m_{a^*}(X_i)\right)^2\,\right|\,I_2\right]\right]$$

$$=\mathbb{E}\left[\mathrm{Var}\left[\left(\left(\frac{1}{l_1}\sum_{i\in I_1}\frac{A_i-g_{1-a^*}(X_i)}{g_{a^*}(X_i)-g_{1-a^*}(X_i)}\frac{Z_i-h_{1-a^*}(X_i)}{h_{a^*}(X_i)-h_{1-a^*}(X_i)}\frac{1}{e_{a^*}(X_i)}-1\right)\big(m_{a^*}(X_i)-\hat{m}_{a^*}(X_i)\big)\right)\,\bigg|\,I_2\right]\right]$$

$$=\frac{1}{l_1}\,\mathbb{E}\left[\mathrm{Var}\left[\left(\left(\frac{A_i-g_{1-a^*}(X_i)}{g_{a^*}(X_i)-g_{1-a^*}(X_i)}\frac{Z_i-h_{1-a^*}(X_i)}{h_{a^*}(X_i)-h_{1-a^*}(X_i)}\frac{1}{e_{a^*}(X_i)}-1\right)\big(m_{a^*}(X_i)-\hat{m}_{a^*}(X_i)\big)\right)\,\bigg|\,I_2\right]\right]$$

$$\leq\frac{O_p(1)}{l_1}\,\sup_{x\in\mathcal{X}}\big|m_{a^*}(X_i)-\hat{m}_{a^*}(X_i)\big|^2\;=\;\frac{o_p(1)}{l}.$$

$$(9)$$

**Analysis of $K_2$.**

Define the term:

$$\left(\frac{A_i-\hat{g}_{1-a^*}(X_i)}{\hat{g}_{a^*}(X_i)-\hat{g}_{1-a^*}(X_i)}\frac{Z_i-\hat{h}_{1-a^*}(X_i)}{\hat{h}_{a^*}(X_i)-\hat{h}_{1-a^*}(X_i)}\frac{1}{\hat{e}_{a^*}(X_i)}\right.$$

$$\left.-\,\frac{A_i-g_{1-a^*}(X_i)}{g_{a^*}(X_i)-g_{1-a^*}(X_i)}\frac{Z_i-h_{1-a^*}(X_i)}{h_{a^*}(X_i)-h_{1-a^*}(X_i)}\frac{1}{e_{a^*}(X_i)}\right)\;\equiv\;(I).$$

We consider the following decomposition:

$$(I) = \left( \frac{A_i - \hat{g}_{1-a^*}(X_i)}{\hat{g}_{a^*}(X_i) - \hat{g}_{1-a^*}(X_i)} \frac{1}{\hat{e}_{a^*}(X_i)} - \frac{A_i - g_{1-a^*}(X_i)}{g_{a^*}(X_i) - g_{1-a^*}(X_i)} \frac{1}{e_{a^*}(X_i)} \right)$$

$$\times \left( \frac{Z_i - \hat{h}_{1-a^*}(X_i)}{\hat{h}_{a^*}(X_i) - \hat{h}_{1-a^*}(X_i)} - \frac{Z_i - h_{1-a^*}(X_i)}{h_{a^*}(X_i) - h_{1-a^*}(X_i)} \right)$$

$$+ \frac{A_i - g_{1-a^*}(X_i)}{g_{a^*}(X_i) - g_{1-a^*}(X_i)} \frac{1}{e_{a^*}(X_i)} \left( \frac{Z_i - \hat{h}_{1-a^*}(X_i)}{\hat{h}_{a^*}(X_i) - \hat{h}_{1-a^*}(X_i)} - \frac{Z_i - h_{1-a^*}(X_i)}{h_{a^*}(X_i) - h_{1-a^*}(X_i)} \right)$$

$$+ \frac{Z_i - h_{1-a^*}(X_i)}{h_{a^*}(X_i) - h_{1-a^*}(X_i)} \left( \frac{A_i - \hat{g}_{1-a^*}(X_i)}{\hat{g}_{a^*}(X_i) - \hat{g}_{1-a^*}(X_i)} \frac{1}{\hat{e}_{a^*}(X_i)} - \frac{A_i - g_{1-a^*}(X_i)}{g_{a^*}(X_i) - g_{1-a^*}(X_i)} \frac{1}{e_{a^*}(X_i)} \right)$$

$$= \left( \frac{A_i - \hat{g}_{1-a^*}(X_i)}{\hat{g}_{a^*}(X_i) - \hat{g}_{1-a^*}(X_i)} - \frac{A_i - g_{1-a^*}(X_i)}{g_{a^*}(X_i) - g_{1-a^*}(X_i)} \right) \left( \frac{Z_i - \hat{h}_{1-a^*}(X_i)}{\hat{h}_{a^*}(X_i) - \hat{h}_{1-a^*}(X_i)} - \frac{Z_i - h_{1-a^*}(X_i)}{h_{a^*}(X_i) - h_{1-a^*}(X_i)} \right)$$

$$\times \left( \frac{1}{\hat{e}_{a^*}(X_i)} - \frac{1}{e_{a^*}(X_i)} \right)$$

$$+ \left( \frac{A_i - \hat{g}_{1-a^*}(X_i)}{\hat{g}_{a^*}(X_i) - \hat{g}_{1-a^*}(X_i)} - \frac{A_i - g_{1-a^*}(X_i)}{g_{a^*}(X_i) - g_{1-a^*}(X_i)} \right)$$

$$\times \left( \frac{Z_i - \hat{h}_{1-a^*}(X_i)}{\hat{h}_{a^*}(X_i) - \hat{h}_{1-a^*}(X_i)} - \frac{Z_i - h_{1-a^*}(X_i)}{h_{a^*}(X_i) - h_{1-a^*}(X_i)} \right) \frac{1}{e_{a^*}(X_i)}$$

$$+ \frac{A_i - g_{1-a^*}(X_i)}{g_{a^*}(X_i) - g_{1-a^*}(X_i)} \left( \frac{Z_i - \hat{h}_{1-a^*}(X_i)}{\hat{h}_{a^*}(X_i) - \hat{h}_{1-a^*}(X_i)} - \frac{Z_i - h_{1-a^*}(X_i)}{h_{a^*}(X_i) - h_{1-a^*}(X_i)} \right) \left( \frac{1}{\hat{e}_{a^*}(X_i)} - \frac{1}{e_{a^*}(X_i)} \right)$$

$$+ \frac{A_i - g_{1-a^*}(X_i)}{g_{a^*}(X_i) - g_{1-a^*}(X_i)} \left( \frac{Z_i - \hat{h}_{1-a^*}(X_i)}{\hat{h}_{a^*}(X_i) - \hat{h}_{1-a^*}(X_i)} - \frac{Z_i - h_{1-a^*}(X_i)}{h_{a^*}(X_i) - h_{1-a^*}(X_i)} \right) \frac{1}{e_{a^*}(X_i)}$$

$$+ \left( \frac{A_i - \hat{g}_{1-a^*}(X_i)}{\hat{g}_{a^*}(X_i) - \hat{g}_{1-a^*}(X_i)} - \frac{A_i - g_{1-a^*}(X_i)}{g_{a^*}(X_i) - g_{1-a^*}(X_i)} \right) \frac{Z_i - h_{1-a^*}(X_i)}{h_{a^*}(X_i) - h_{1-a^*}(X_i)} \left( \frac{1}{\hat{e}_{a^*}(X_i)} - \frac{1}{e_{a^*}(X_i)} \right)$$

$$+ \left( \frac{A_i - \hat{g}_{1-a^*}(X_i)}{\hat{g}_{a^*}(X_i) - \hat{g}_{1-a^*}(X_i)} - \frac{A_i - g_{1-a^*}(X_i)}{g_{a^*}(X_i) - g_{1-a^*}(X_i)} \right) \frac{Z_i - h_{1-a^*}(X_i)}{h_{a^*}(X_i) - h_{1-a^*}(X_i)} \frac{1}{e_{a^*}(X_i)}$$

$$+ \frac{Z_i - h_{1-a^*}(X_i)}{h_{a^*}(X_i) - h_{1-a^*}(X_i)} \frac{A_i - g_{1-a^*}(X_i)}{g_{a^*}(X_i) - g_{1-a^*}(X_i)} \left( \frac{1}{\hat{e}_{a^*}(X_i)} - \frac{1}{e_{a^*}(X_i)} \right). \tag{10}$$

Define:

$$\Delta g_i := \frac{A_i - g_{1-a^*}(X_i)}{g_{a^*}(X_i) - g_{1-a^*}(X_i)}, \qquad \Delta \hat{g}_i := \frac{A_i - \hat{g}_{1-a^*}(X_i)}{\hat{g}_{a^*}(X_i) - \hat{g}_{1-a^*}(X_i)},$$

$$\Delta h_i := \frac{Z_i - h_{1-a^*}(X_i)}{h_{a^*}(X_i) - h_{1-a^*}(X_i)}, \qquad \Delta \hat{h}_i := \frac{Z_i - \hat{h}_{1-a^*}(X_i)}{\hat{h}_{a^*}(X_i) - \hat{h}_{1-a^*}(X_i)},$$

$$\Delta e_i := \frac{1}{e_{a^*}(X_i)}, \qquad \Delta \hat{e}_i := \frac{1}{\hat{e}_{a^*}(X_i)}.$$

Therefore, combining equations equation 10 and part $K_2$, we separate $K_2$ into five mean-zero terms and two product terms. By triply robustness of equation 4, since $g_{a^*}(x)$ and $m_{a^*}(x)$ are correctly specified:

$$\mathbb{E}\left[ \Delta g_i (\Delta \hat{h}_i - \Delta h_i)(Y_i - m_{a^*}(X_i))\Delta e_i \right] = 0, \tag{11}$$

Since $h_{a^*}(x)$ and $m_{a^*}(x)$ are correctly specified:

$$\mathbb{E}\Big[(\Delta \hat{g}_i - \Delta g_i)\, \Delta h_i\, (\Delta \hat{e}_i - \Delta e_i)(Y_i - m_{a^*}(X_i))\Big] = 0, \tag{12}$$

The correct specification of $h_{a^*}(x)$ and $m_{a^*}(x)$ leads to:

$$\mathbb{E}\Big[(\Delta \hat{g}_i - \Delta g_i)\, \Delta h_i\, \Delta e_i\, (Y_i - m_{a^*}(X_i))\Big] = 0, \tag{13}$$

Similarly, since $g_{a^*}(x)$ and $m_{a^*}(x)$ are correctly specified:

$$\mathbb{E}\Big[\Delta h_i\, \Delta g_i\, (\Delta \hat{e}_i - \Delta e_i)(Y_i - m_{a^*}(X_i))\Big] = 0, \tag{14}$$

Also $g_{a^*}(x)$ and $m_{a^*}(x)$ are correctly specified:

$$\mathbb{E}\Big[(\Delta h_i - \Delta \hat{h}_i)\, \Delta g_i\, (\Delta \hat{e}_i - \Delta e_i)(Y_i - m_{a^*}(X_i))\Big] = 0, \tag{15}$$

In addition, by the Cauchy-Schwarz inequality, we have

$$\frac{1}{l_1}\sum_{i \in I_1}(\Delta \hat{g}_i - \Delta g_i)(\Delta \hat{h}_i - \Delta h_i)(Y_i - m_{a^*}(X_i))\Delta e_i$$

$$\leq \sqrt{\frac{1}{l_1}\sum_{i \in I_1}(Y_i - m_{a^*}(X_i))\Delta e_i(\Delta \hat{g}_i - \Delta g_i)^2}\sqrt{\frac{1}{l_1}\sum_{i \in I_1}(Y_i - m_{a^*}(X_i))\Delta e_i(\Delta \hat{h}_i - \Delta h_i)^2}$$

$$= o_p(g_n h_n). \tag{16}$$

With the same logic, we have

$$\frac{1}{l_1}\sum_{i \in I_1}(\Delta \hat{g}_i - \Delta g_i)(\Delta \hat{h}_i - \Delta h_i)(\Delta \hat{e}_i - \Delta e_i)(Y_i - m_{a^*}(X_i)) = o_p(g_n h_n e_n) \tag{17}$$

Therefore, combining equations equation 11, equation 12, equation 13, equation 14, equation 15, equation 16, equation 17, we have that

$$K_2 = o_p(\max(g_n h_n, g_n h_n e_n)) = o_p(g_n h_n). \tag{18}$$

**Analysis of $K_3$.**

Combine equations equation 10 and the same decomposition of $(I)$ in $K_3$, we can get $K_3 = o_p(\max(m_n g_n, m_n h_n, m_n e_n))$ with same decomposition process. So we have that $K_1 + K_2 + K_3 = o_p(\max(m_n g_n, m_n h_n, m_n e_n, g_n h_n))$, which means $\hat{\mu}_{a^*}^{I_1}(x) - \tilde{\mu}_{a^*}^{I_1}(x) = o_p(\max(m_n g_n, m_n h_n, m_n e_n, g_n h_n))$, and therefore $\hat{\mu}_{a^*}(x) - \tilde{\mu}_{a^*}(x) = o_p(\max(m_n g_n, m_n h_n, m_n e_n, g_n h_n))$, so it's obvious to get

$$\hat{\tau}(x) - \tilde{\tau}(x) = o_p(\max(m_n g_n, m_n h_n, m_n e_n, g_n h_n)),$$

which completes the proof.

$$K_3 = o_p\big(\max(m_n g_n, m_n h_n, m_n e_n)\big),$$
$$K_1 + K_2 + K_3 = o_p\big(\max(m_n g_n, m_n h_n, m_n e_n, g_n h_n)\big),$$
$$\hat{\mu}_{a^*}^{I_1}(x) - \tilde{\mu}_{a^*}^{I_1}(x) = o_p\big(\max(m_n g_n, m_n h_n, m_n e_n, g_n h_n)\big),$$
$$\hat{\mu}_{a^*}(x) - \tilde{\mu}_{a^*}(x) = o_p\big(\max(m_n g_n, m_n h_n, m_n e_n, g_n h_n)\big),$$
$$\hat{\tau}(x) - \tilde{\tau}(x) = o_p\big(\max(m_n g_n, m_n h_n, m_n e_n, g_n h_n)\big).$$

## D    PROOF OF THEOREM 3.8

Using the forest weights $\alpha_i(x)$, which form the basis of the generalized random forest estimator $\tilde{\tau}^*(x)$ under unknown nuisance parameters, we obtain the following linear approximation:

$$\tilde{\tau}^*(x) = \tau(x) + \sum_{i=1}^{n} \alpha_i(x)\rho_i^*(x),$$

where $\rho_i^*(x)$ denotes the influence function for the $i$-th observation with respect to the true parameter value $\tau^*(x)$, and $\tilde{\tau}^*(x)$ is the pseudoforest output defined by weights $\alpha_i(x)$ and outcome $\tau(x) + \rho_i^*(x)$.

Assumptions 2–6 from Athey et al. (2019) are satisfied by construction of the estimating equation $\psi_{\tau(x)}$. In particular, $\psi_{\tau(x)}(x)$ is Lipschitz continuous in $\tau(x)$ (satisfying their Assumption 4), and the solution to $\sum_{i=1}^{n} \alpha_i \psi_{\tau(x)}^i = 0$ always exists (satisfying their Assumption 5). From Wager & Athey (2018), there exists a sequence $\sigma_n(x)$ such that

$$[\tilde{\tau}^*(x) - \tau(x)]/\sigma_n(x) \to N(0, 1),$$

where $\sigma_n^2(x) = polylog(n/s)^{-1}s/n$, and $polylog(n/s)$ is bounded away from zero and grows at most polynomially in $\log(n/s)$.

Furthermore, by Lemma 4 in Athey et al. (2019), we have

$$(n/s)^{1/2}[\tilde{\tau}(x) - \tilde{\tau}^*(x)] = O_p\left(\max\left(s^{-\frac{\pi \log((1-\nu)^{-1})}{2\log(\nu^{-1})}}, (s/n)^{1/6}\right)\right).$$

Finally, applying Lemma 3.6, as long as

$$o_p\big(\max(m_n g_n, m_n h_n, m_n e_n, g_n h_n)\big)$$

decays faster than $polylog(n/s)^{-1/2}(s/n)^{1/2}$, we obtain

$$[\hat{\tau}(x) - \tau(x)]/\sigma_n(x) \to N(0, 1).$$

## E    SIMULATION DATA GENERATING PROCESS

We follow the setup in Zhou & Tchetgen Tchetgen (2024). Covariates $X = (X_1, X_2)^\top$ are drawn from a multivariate Gaussian distribution $N(\mathbf{0}, I_2)$. Conditional on $X$, the binary treatment $A^*$ is distributed as

$$\Pr(A^* = 1 \mid X) = \frac{1}{1 + \exp(-X_1) + \exp(-X_2)}.$$

Surrogate $A$ and proxy $Z$ are generated as:

$$\Pr(A = 1 \mid A^* = 1, X) = \text{logit}^{-1}(1 + 2X_1),$$
$$\Pr(A = 1 \mid A^* = 0, X) = \text{logit}^{-1}(-1 + 0.5X_1 - X_2),$$
$$\Pr(Z = 1 \mid A^* = 1, X) = \text{logit}^{-1}(-1 - X_1 + 0.5X_2),$$
$$\Pr(Z = 1 \mid A^* = 0, X) = \text{logit}^{-1}(2 + X_1),$$

where $\text{logit}(p) = \ln\{p(1-p)^{-1}\}$.

The outcome $Y$ is considered in two settings, binary and continuous, incorporating treatment heterogeneity:

$$\text{i. } \Pr(Y = 1 \mid A^* = 1, X) = \text{logit}^{-1}(-2 + 0.5X_1 + X_2),$$
$$\Pr(Y = 1 \mid A^* = 0, X) = \text{logit}^{-1}(1 - 2X_1 + 0.5X_2);$$

$$\text{ii. } Y = \sin(\pi X_1) + (A^* - 0.5)(X_1 + X_2) + \varepsilon, \quad \varepsilon \sim \mathcal{N}(0, 1).$$

The default tuning parameters from the **grf** package were used for all forest-based methods: 2000 trees, minimum node size of 5, subsampling fraction of 0.5, and number of split variables equal to $p$ (the number of confounders). In Figure 2a, all forests were trained on a training set of size $n_{\text{train}} = 5000$ and evaluated on a test set of size $n_{\text{test}} = 5000$.

## F  REAL DATA

Since treatment effects are unobservable in real-world data, the evaluation of effect estimators is inherently challenging due to the absence of ground truth. This precludes the use of standard error metrics. To address this, Yadlowsky et al. (2024) proposed a family of evaluation metrics for treatment prioritization rules, known as rank-weighted average treatment effects (RATE).

The central object in RATE is the calibration curve, or Targeting Operator Characteristic (TOC), defined as

$$\text{TOC}(q) = \mathbb{E}\Big[Y_i(1) - Y_i(0) \mid \hat{\tau}(X_i) \geq F^{-1}_{\hat{\tau}(X_i)}(1 - q)\Big] - \mathbb{E}[Y_i(1) - Y_i(0)].$$

This curve ranks all observations by $\hat{\tau}(x)$ and compares the ATE among the top $q$ fraction of prioritized units with the population ATE. The RATE metric is the area under the TOC curve: higher values indicate that the CATE estimator effectively stratifies the population by treatment heterogeneity.

Figure 3 presents the TOC curve for all observations estimated by LTCF. The RATE, computed as the area under this curve, is 233.59 mm$^3$ with a bootstrapped standard error of 13.13. This suggests substantial heterogeneity in response to AD status. For instance, patients in the lowest 20th quantile of estimated CATEs exhibit hippocampal volumes, on average, 400 mm$^3$ lower than the population mean under AD status, indicating significantly greater adverse effects in this subgroup.

## LARGE LANGUAGE MODELS USAGE

The authors used large language models (LLMs) solely for grammar checking. All responsibility for the content remains with the authors.

