# OpenReview forum: "Latent Causal Forest:  Estimating Heterogeneous Treatment Effects under a Hidden Treatment"
_ICLR.cc/2026/Conference — Submitted to ICLR 2026_

### Official Review · Reviewer_fSBC · 2025-10-27

**Soundness:** 3
**Presentation:** 3
**Contribution:** 2
**Rating:** 2
**Confidence:** 4

**Summary:**

The paper studies estimation of conditional average treatment effects (CATE) when the true binary treatment $A^{\star}$ is unobserved and only a misclassified surrogate $A$ and a proxy $Z$ are observed. It follows the identification framework of Zhou and Tchetgen Tchetgen (2024), under which an efficient influence function (EIF) score for the hidden-treatment setting is available. The paper also adopts their approach to estimate the nuisance functions $e_a(x)=\operatorname{Pr}\left(A^{\star}=a \mid X=x\right), m_a(x)=\mathbb{E}\left[Y \mid A^{\star}=a, X=x\right]$, $g_a(x)=\mathbb{E}\left[A \mid A^{\star}=a, X=x\right]$, and $h_a(x)=\mathbb{E}\left[Z \mid A^{\star}=a, X=x\right]$. The main contribution is Latent Causal Forest (LTCF), which extends generalized/causal forests to the unobserved-treatment case by using forest weights to localize the EIF score and estimate $\tau(x)$. Theoretical results establish consistency and asymptotic normality under standard honest-forest conditions and reasonable rate requirements on the nuisance estimators. The paper also presents a two-stage DR-learner variant for hidden treatment. In simulations, LTCF lowers mean squared error relative to forests that treat the surrogate or the proxy as the treatment. A case study on ADNI uses MMSE subscores as the surrogate and proxy and estimates the effect of latent Alzheimer's disease status on hippocampal volume.

**Strengths:**

The paper addresses CATE with a hidden binary treatment. Its identification assumptions are the same as Zhou and Tchetgen Tchetgen (2024), so the conceptual novelty on identification is limited. The main value lies elsewhere: the authors turn that identification into a practical learner that uses an influence-function score, triply robust structure, honest splitting, and forest-based weighting. This yields a clear algorithm with end-to-end training and inference for heterogeneous effects, which prior work focused on average effects did not operationalize at this level. The empirical section is careful and shows consistent gains over naïve baselines. The writing states assumptions in a transparent way and makes the estimation steps concrete. In short, while the identification itself follows known conditions, the paper contributes by engineering a usable pipeline for hidden-treatment CATE with sound inference, which can matter for applied fields where heterogeneity is critical.

**Weaknesses:**

1. A major weakness is originality relative to Zhou and Tchetgen Tchetgen (2024). The paper adopts the same identification assumptions. While Zhou and Tchetgen Tchetgen (2024) target the ATE, extending those conditions to CATE in the binary proxy setting requires little extra work, so the identification for CATE does not appear new. The abstract and introduction also read as if this were the first framework for CATE with a hidden binary treatment. Please justify this claim by placing the work in the context of prior hidden-treatment papers that focus on ATE, and explain what is new beyond moving from ATE to CATE. Relatedly, Section 3.5 appears to construct a DR-learner by combining the EIF in Zhou and Tchetgen Tchetgen (2024) with the DR-learner framework of Kennedy (2023). If so, please state this clearly and identify which parts are new, and which parts are direct adaptations. In addition, explain why generalized random forests applied to pseudo-outcomes imputed via the EM algorithm (Algorithm 1 in Zhou and Tchetgen Tchetgen, 2024) is not a natural and competitive baseline. A side-by-side comparison, using the same cross-fitting and tuning budget, would make the design choice transparent.

2. Scope of the surrogate/proxy construction. Identification and proofs are written for binary $A, Z$ ( $2 \times 2$ matrices in Appendix B). The main text sometimes reads as if $Z$ could be general. Please state early that the formal results require binary $Z$, or provide the extension (and conditions) for multi-level or continuous proxies/surrogates.

3. Theory: rate conditions and guidance. The main CLT requires the product of nuisance errors to vanish faster than a polylog factor tied to subsample size $s$ (Theorem 3.8). This is hard to interpret in practice. Can the authors give concrete sufficient conditions under common nonparametric rates (e.g., Hölder smooth $m_{a}$ with random forests or kernels) and translate the bound into sample-size guidance? Also spell out recommended $s$, honesty parameters, and the "little bags" settings used in experiments.

4. Why forests? The paper does not explain why a forest is the right base learner given many viable ML choices. Please add motivation and evidence.

**Questions:**

1. Practical rates. Theorem 3.8 imposes a product-rate condition on $m, g, h, e$. Can you give a concrete example with kernel estimators (bandwidth $h_n$ ) or forests (leaf size, honesty) that satisfies this condition, and translate it into a rule-of-thumb for $n$ and tree parameters?

2. Inference details. The paper uses the “bootstrap of little bags,” but gives little detail. Please add a clear primer on the method (e.g., what is resampled, how bag size and the number of bags are set, and the runtime cos).

3. Baselines. Could you add an EM-impute-then-GRF baseline to show the specific gain from the triply robust score and forest weighting?

---

### Official Review · Reviewer_hrBY · 2025-10-30

**Soundness:** 3
**Presentation:** 2
**Contribution:** 2
**Rating:** 2
**Confidence:** 3

**Summary:**

The paper introduces a new forest-based algorithm for estimating CATE under the setting where the (binary) treatment cannot be observed. The authors presented theories and empirical results to validate their proposal.

**Strengths:**

The paper gives a very detailed description of the preliminaries of causal inference and related tools (including influence functions) that are used in the paper.

The paper tackles a problem similar to proximal causal inference, which is an important field of study in the recent era due to the prevalence of observational data with measurement errors or censoring issues.

The description of the technical methodologies of the paper is concise and structurally clear.

**Weaknesses:**

Misleading organization: the name of section 2, "Problem Statement", is confusing, and should be named "Preliminaries" or "Background" instead, as it introduces the background knowledge of causal inference rather than the actual objective that the paper aims to tackle. The objective of the paper first appeared in section 3.1, which should be named "Problem Statement" instead.

Lack of novelty: Many of the methods and theories of the paper seem to rely heavily on the prior work of Zhou and Tchetgen Tchetgen (2024). The main components of the proposed algorithm seem to be a combination of existing works and methods. There does not seem to be any significant technical innovations that need to be introduced to extend their work in the conditional setting, which was claimed to be "first achieved" by this paper among existing literature. This could be an overclaim statement.

Lack of comparison with existing settings: Given the presumed lack of novelty, I find it necessary to include a section that details the differences in the setting that the work aims to address, which is not included in the current version of the paper. Specifically, in addition to the verbal description of the related works, the authors should also mathematically/visually show how the assumed causal DAG studied in this paper differs from existing settings, such as instrumental variables and proximal causal inference in general.

Unclear purpose of introducing the DR-learner: the authors introduced a variant of their proposed algorithm using the DR-learner as their base model. This proposal was never compared to the previous random forest approach, and no insights have been drawn on how practitioners should choose between these alternatives.

Weak experiment: The experiment was done entirely with ablation models of the proposed methods in this paper. No comparison with external models as baselines was presented (e.g., proximal causal forests, latent causal inference models, or confounding causal inference models)

Missing derivations: the derivation of (1) (4) and (7) being the score function, and how they satisfy the doubly/triply robust properties were not provided in the paper.

**Questions:**

What is the difference between the problem setting studied in this work compared to proximal causal inference, causal inference with unobserved confounders of the treatments, and causal inference with measurement error?

What novelties or techniques in the theoretical proofs did the paper first introduce? What is the main technical challenge for considering estimating CATE in your setting?

Where is the proof for (1) (4), and (7) as the derived score function, and where is the proof for their robustness guarantees?

---

### Official Review · Reviewer_sjMP · 2025-10-31

**Soundness:** 3
**Presentation:** 3
**Contribution:** 2
**Rating:** 6
**Confidence:** 3

**Summary:**

This paper introduces the Latent Causal Forest (LTCF), a new framework for estimating conditional average treatment effects (CATE) when the true treatment is unobserved or misclassified. Building upon generalized random forests, the authors incorporate surrogate and proxy variables to recover identification under hidden treatment status, supported by semiparametric efficiency theory. The proposed EIF-based estimator enjoys triple robustness, ensuring consistent estimation if any subset of nuisance components is correctly specified. The paper also extends the DR-Learner framework to the hidden-treatment setting, providing a flexible two-stage alternative. Through comprehensive simulations, the method shows strong accuracy and type-I error control compared with surrogate-based causal forests. A real-world application on the ADNI dataset demonstrates meaningful subgroup heterogeneity in Alzheimer’s disease effects on hippocampal volume, confirming the method’s practical relevance.

**Strengths:**

1. The paper addresses an important and underexplored problem — estimating heterogeneous treatment effects when the true treatment variable is unobserved or misclassified — which is highly relevant to real-world causal inference tasks.

2. The proposed Latent Causal Forest (LTCF) extends generalized random forests to the hidden-treatment setting and introduces an EIF-based triply robust estimator, representing a theoretically grounded and flexible contribution.

3. The paper provides clear identification assumptions and derives consistency and asymptotic normality results under hidden treatments, enhancing the methodological rigor.

4. Both synthetic and real-world experiments (ADNI dataset) are carefully designed to validate the framework, showing substantial performance gains over surrogate or proxy-based baselines.

**Weaknesses:**

1. The identification of hidden treatment effects relies heavily on conditional independence between the surrogate, proxy, and outcome variables, which may be unrealistic or empirically unverifiable

2. Although the estimator is claimed to be triply robust, the simulations do not test partial misspecification cases (e.g., when one nuisance model is wrong), leaving the empirical robustness claim insufficiently supported.

3. The experimental evaluation lacks baselines against recent proximal causal inference or proxy-based CATE estimators or even some existing models for observed treatment setting (such as CFR, DR-CFR etc).

4. identifying Z is quite important but hard to find, how to address this issue if wo do not know what variable can be viewed as "Z"? It seems this paper lacks of the discussion about how to identify Z.

**Questions:**

The following is my concerns and questions:

1. The motivation of this paper is unclear in the intro. It mentions the measurement error settings but it seems the goal is to estimate HTE under latent (unobserved) treatment. So, the setting is to infer or recover the latent treatment or correct the measuring error of the observed treatment? This is unclear.

2. The equation of latent outcome bias are unclear. What is F(x) here in section 3.1?

3. Assumption 3.2 imposes strong conditional independence among Yi,Ai​, and Z_i​ given A_i^*​ and Xi​. This may rarely hold in practice—for instance, if Zi affects both Ai and Yi through confounding or measurement error channels. The authors should discuss its empirical plausibility and potential robustness checks.

4. In the assumption 3.3, these conditions are essentially designed to prevent completely random misclassification right? however, in many practical domains, the misclassification is often random (e.g., random incorrectly recorded data due to to the measurement). Furthermore,
Assumption 3.3 prevents complete random misclassification but does not guarantee numerical stability when the misclassification rate is near 0.5. The identification might be weak, leading to high variance or non-robust CATE estimates.

5. While Assumptions 3.1–3.2 suggest that Z is correlated with A∗ but conditionally independent of Y, the paper does not clarify what kind of proxy Z satisfies this in real data. If Z is measured from the same process as A, the independence assumption could easily fail.

6. The method claims to retain triply robust properties, yet DR-learner theory (Kennedy, 2023) generally ensures only double robustness. It remains unclear how triple robustness is preserved after pseudo-outcome regression, and whether additional conditions (e.g., independence of proxies, low noise) are required.

7. Cross-fitting is mentioned to ensure robustness, yet the paper omits key implementation details (number of folds, splitting strategy, and whether (A,Z) models are cross-fitted jointly). Asymmetric cross-fitting could lead to residual overfitting bias.

8. Although the DR-learner is claimed to be triply robust, the experiments do not include scenarios where one or two nuisance models are misspecified. Without such stress tests, the empirical value of triple robustness remains unclear.

---

### Official Review · Reviewer_ab7n · 2025-10-31

**Soundness:** 4
**Presentation:** 3
**Contribution:** 2
**Rating:** 6
**Confidence:** 4

**Summary:**

In recent work, Zhou & Tchetgen Tchetgen (2024) (hereon ZT) established identification of certain aggregate causal quantities, notably the average treatment effect (ATE), when the true binary treatment $A^{*}$ is observed only with some error. To achieve this, a proxy $Z$ for the true underlying treatment is required in addition to $A$ (the imperfect measurement of the true treatment), and a number of a priori assumptions must be made regarding the data generating process. The present paper applies the ideas in that work to the problem of conditional average treatment effect (CATE) estimation with a mismeasured treatment. The authors show that the generalized random forest (GRF) method for CATE estimation proposed by Athey et al. (2019) can be combined with the approach of ZT.

In the standard random forest algorithm splits in individual regression trees are chosen to minimize within-leaf mean-squared prediction errors (regularization and random selection of splitting variables aside). By contrast, the splits in decision trees in the GRF are grown to maximize the between-leaf variance in leaf-specific ATE estimates. In addition, the final CATE estimates are given by an average (over trees) of within-leaf ATE estimates. Thus, by replacing the usual ATE estimates with those that account for mismeasurement of the treatment variable, CATE estimates can be obtained from the GRF that are consistent when treatment is mismeasured. My description here is a bit of a simplification, for example, in the tree growing process the authors use the between-leaf variation in the average score function rather than the ATEs for computational expediency, but the idea is much the same.

The authors show that their CATE estimation method is consistent and asymptotically centered normal under some high-level conditions on the convergence rates of the nuisance parameter estimates. They provide some simulation evidence of its efficacy and apply the method to real data (namely to data from the Alzheimer’s Disease Neuroimaging Initiative, the same application as in ZT).

The authors also briefly describe an approach based on the DR-learner of Kennedy (2023) rather than GRFs.

**Strengths:**

I think the paper makes a meaningful contribution. I believe the extension of ZT's analysis to CATE estimation could prove useful to applied researchers. The paper is quite clear and generally well-written. The results appear to be correct and the asymptotic analysis is sufficiently thorough.

**Weaknesses:**

The paper discusses semiparametric efficiency a fair bit, however CATE estimation is a non-parametric problem, and so I am not sure how relevant this is to the problem at hand. For example, I do not know whether this CATE estimator achieves an optimal convergence rate under the stated assumptions. Similarly, while the estimator is doubly robust to many of the nuisance parameters, it is not doubly robust with respect to the random forest weights. I wonder if more could be said about this, presumably the same questions arise in the context of generalized random forests more broadly, which are usually employed using doubly robust score functions in other contexts.

I think the paper could be a bit clearer about what is original relative to ZT. In particular, Lemma 3.4 appears to be only a very minor adaptation of ZT, yet the present work contains a self-contained proof (which incidentally glosses over the key step of using Assumption 3.3 to order the eigenvalues).

The asymptotic results also rely on quite high-level assumptions about the convergence rates of nuisance parameter estimates. This is not inherently a problem, but it might be good to cite some work readers could look at if they wish to find specific rates for the nuisance parameter estimates under primitive conditions.

**Questions:**

How do the results in the Alzheimer's Disease application compare to those in the existing literature?

---

### Meta-Review · Area_Chair_ePFs · 2025-12-23

**Summary:**

Paper proposes a multiply robust framework for conditional average treatment effect estimation with latent treatments under the presence of proxies (treatments and other variables). Under conditions of relevance, and noise level between observed treatment A and true treatment A*, identifiability of CATE is demonstrated. Eigenvalue decomposition can be used to estimate outcome models under hidden treatment. A multiply robust estimator is presented based on this identifiability result. Causal forests are extended to estimate the average treatment effect estimator. Theoretical consistency is demonstrated. Experimental evaluation includes synthetic simulation and results on ADNI dataset (alzheimer's disease progression).

Reviewers cite prior work from Zhou & Tchetgen Tchetgen which focuses on ATE estimation with hidden treatments and/or noisy measurements of treatment assignments as the main building block that is adapted in this work for CATE.

Reviewers pointed out that overall contribution is well written and well presented, except not contextualized well enough in prior work.

Robustness with respect to parameters of the causal forests is not discussed.

Empirical discussions about how Z should be selected are lacking

Some lack of clarity on assumptions in context of prior work is also missing

**Reviewer Concerns:**

No rebuttal was submitted, all concerns remain outstanding

**Reviewer Scores:**

No update to scores expected because of missing rebuttal

---

### Decision · Program_Chairs · 2026-01-26

Reject